# Privacy-Preserving Data Sharing in Cloud IoT on Solid[*]

Fatmah Mashat[1], Maribel Fernandez[1]

[1]*Department of Informatics, King's College London, London, UK*

**Abstract**

The continuous growth of Smart Home IoT has increased the complexity of governing personal data. This paper presents a category-based approach to simplify the management of user privacy policies within a Databank architecture built upon Solid pods. We present a layered architecture that separates data collection, policy reasoning, automated data transfer, and storage in Solid pods. Two access control enforcement models are analysed: a Solid-mediated model, which depends on Solid's built-in access control mechanisms and is restricted to static policies, and a Databank-mediated model, which uses Solid only for storage and enforces category-based rules externally. This shows that the same category policy can support different trust and enforcement distributions, thus integrating category-based control with Solid offers a flexible and explainable basis for user-centric smart-home data management.

**Keywords**

Category-Based Access Control, IoT Privacy, Databank, Solid, Data sharing, policy preference

## 1. Introduction

The Internet of Things (IoT) in smart home environments produces, collects, shares and processes continuous streams of personal data via devices, like sensors, cameras, and smart door locks. While some of the data that is collected and processed is essential for the functionality of the IoT devices, a substantial amount is not strictly required. This introduces new challenges regarding user privacy and data protection due to excessive data collection. Privacy policies are often lengthy and complex, even though their purpose is to help the user understand the usage of their data. Users (especially non-technical ones) struggle to understand data usage [1]. Many services are offered for free, but in exchange for users' personal data, turning the data into a payment method [1]. Therefore, it is essential for users to have mechanisms that will help them understand and manage the use of their data. However, capturing user preferences and turning them into enforceable policies is a major challenge [1]. Some studies have looked into structured annotation and information extraction techniques that identify key attributes to make policies more transparent [1]. Existing access control models provide detailed specification mechanisms to control access to resources, but they were not developed to deal with data sharing and user-centric usability. The Smart Home Databank [2] was created as a privacy-preserving architecture that places smart-home data into a user-controlled repository and controls access to that data through category-based policies. This paper proposes a category-based data sharing framework integrated into a Databank architecture that employs Solid decentralised storage to make IoT privacy management more explainable and user-centred.

## 2. Background

### 2.1. Access control

Access control is an essential concept in security. Access control policies define which subject can perform which action on resources under which condition [3].

---

*4th Solid Symposium, 4th Privacy and Personal Data Management Session, 30 April–1 May 2026, London, UK*

[*]*Corresponding author.*

✉ famashatt@gmail.com (F. Mashat); maribel.fernandez@kcl.ac.uk (M. Fernandez)

### 2.1.1. Role-based access control (RBAC) and Attribute-based access control (ABAC)

In RBAC authorisation is centred around roles, whether it is a job function or responsibility. Users will be given a predefined role, and access will be based on the permissions linked to the role [4, 5]. This simplifies administration because roles change less frequently than users, allowing administrators to manage role memberships rather than individual permissions [4]. Extensions for IoT environments, such as the extended generalised RBAC (EGRBAC) [6] adapt RBAC concepts to smart home devices; users (e.g., parents) are assigned roles, and each device has an operation-level permission. These permissions are organised into sets that represent different access levels. Permissions are given through a positive authorisation with constraints preventing unauthorised actions. Despite RBAC stability and ease of management, it lacks flexibility in dynamic situations where permissions must change according to situational or contextual elements.

Unlike RBAC, ABAC [5] bases authorisation on the attributes of the subject, resources, actions and environment rather than fixed roles, so permissions change automatically when attribute values are altered, providing context-aware and dynamic authorisation [3]. It offers fine-grained control at the level of individual operations. However, it can create complexity as the number of attributes increases, as policies can become more challenging to evaluate and create privacy problems, as attributes frequently include personal information [5]. Hybrid Attribute-Based Access Control (HABAC) extends ABAC with relationship attributes that act like roles (e.g., parent, child or guest) while keeping the core logic attribute-based. It also uses dynamic attributes such as time or location, enabling context-aware authorisation [7]. It fills the gap between RBAC manageability and ABAC expressiveness.

### 2.1.2. Category-Based Access Control (CBAC)

The CBAC (Category-Based Access Control) model [8, 9] is an axiomatic framework for the definition of policies based on the idea of categorisation as a mechanism to specify authorisations. Access control policies in the CBAC model specify principal and resource categories and assign permissions (consisting of an action on a resource category) to relevant categories of principals. Categories can be defined in different ways, for example, they can be associated with roles as in RBAC, or defined via attributes of users, resources and the environment, as in ABAC models.

The SHoCBAC [10] model is a category-based access control model for smart homes that permits the definition of RBAC-style and ABAC-style policies as well as a whole spectrum of hybrid RBAC-ABAC policies through user and device categorisation. It leverages cognitive science principles [11, 12] that indicate that categorisation is one of the key mechanisms used to organise knowledge. It focuses on access control for smart home devices. Instead, this paper focuses on securing the data generated by those devices, requiring an architectural approach to data protection and restricted sharing.

## 2.2. DataBank

DataBank [13] is a four-layer cloud-IoT architecture that offers mechanisms to manage data collection at IoT-device level and data sharing at the cloud level. To control data sharing operations, DataBank uses an instance of the Category-Based model, called CBDA (Category-Based Data Access) [14]: policies define the relevant IoT devices and data streams and the applications (also called services) that can access them and specify categories of data and applications so that only data in specific categories can be transferred to the cloud and accessed by relevant categories of services. The topmost Application Layer in DataBank includes an interface for data owners to set up privacy preferences and a data-sharing manager API, which handles service requests. The Cloud Layer includes a Data Access Control Enforcement Module (to filter service requests), an Auditing Module (to log transactions), a Cloud Repository (storage), a Privacy-Utility Mechanism and a Data Sharing Policy. The Data Pocket Layer connects with the IoT devices and has a small memory for data storage before policy-based filtering and transfer to the cloud. Finally, the bottom layer deals with the devices connected to the DataBank.

An instance of DataBank specifically tailored to the Smart Home scenario is described in [2]. In this instance, data sharing is controlled using a CBDASH policy where data streams are categorised on

the basis of the sensitivity of the data (with four basic categories of data: top-secret, secret, private, and public) and services are categorised on the basis of trust (with three basic categories of services: basic, medium, and premium). More categories can be defined using dynamic attributes such as time or location.

## 2.3. Solid Platform

Solid [15] is a decentralised data management framework. It was built to give users ownership and control of their data. They can store and manage their data in Personal Online Data Stores (Pods), independently of the applications, which can be hosted by a third party or by themselves [16]. Pods store structured and unstructured data as containers and resources, allowing fine-grained authorisation at the level of individual resources. Data portability is made possible by this architectural division between storage and applications, by allowing multiple applications to use the same underlying data. It is built on decentralised identity and authentication using WebIDs, which are HTTP IRIs that uniquely identify agents. They are used as the primary identifier for users and application [17]. When a WebID is dereferenced, it returns a RDF document that describes that agent, which is then utilised in authentication and access control [17]. Solid has two access control mechanisms: Web Access Control WAC relies on access control lists (ACLs), which are RDF documents containing authorisation that links agents to access modes such as Read, Write, Append, and Control on resources or containers [18]. When a request is made to access a protected resource, the server finds its associated ACL resource and evaluates the authorisation to grant or deny [18]. Because of its coupling between policies and resources this limits the policy reuse and increases complexity in large-scale environments. Access Control Policy (ACP) was introduced to increase expressiveness and modularity. Policy definitions are separate from the resources, allowing the same policies to be applied to numerous resources and changed independently [19]. Policies can define grant and deny decisions over access modes, as well as combine conditions on agents, client applications, and other attributes, with the pod server enforcing them at request time, making it a more adaptable access control.

## 3. Databank for Smart Homes (with CBDASH on Solid)

The proposed layered architecture is made up of four main layers: IoT devices that produce data, a local hub and Data Pocket that filters data before transferring to the cloud, Solid pods that offer cloud data storage and access control, and application services that access authorised data. The Smart Home Databank acts as an interface between the local smart home devices and Solid cloud, it unites unprocessed data from devices of different communication protocols and reporting rates. All the IoT devices are connected to a local Databank Hub deployed in the smart home, data is collected at the hub and is filtered according to the CBDASH [2] policy into a set of categories (top-secret, secret, private, and public). Filtering and categorisation are performed locally, reducing the amount of data sent to the cloud. The hub contains a Data Pocket that provides temporary storage with a limited capacity, retaining filtered data for up to 24 hours, minimising storage usage and avoiding unnecessary long-term data accumulation. Only data in the secret, private and public categories is transferred to Solid pods, providing long-term storage and access control. More precisely, the hub layer has an application that transfers data streams from the hub to Solid pods, as the immediate filtering and classification of data at the edge enables the smart home to never lose sensitive information from being exposed to the cloud, and so the data is not to be removed from the house prior to policy enforcement being made. As a result, an immediate filtering and classification of data at the hub-based strategy provides great privacy guarantees and successful execution of the policy. The CBDASH categories [2] are important during data organisation at Solid pods. Each category is placed in a separatepodd or a folder in a pod, which gives a logical distinction of data between public, private, and anonymised data. A Public Pod is a storage that holds information that is supposed to be shared publicly. An Encrypted Private Pod is where the encrypted sensitive information is stored. Only authorised applications can decipher it when they retrieve the data. The information can also be divided into service-specific folders in a pod. For

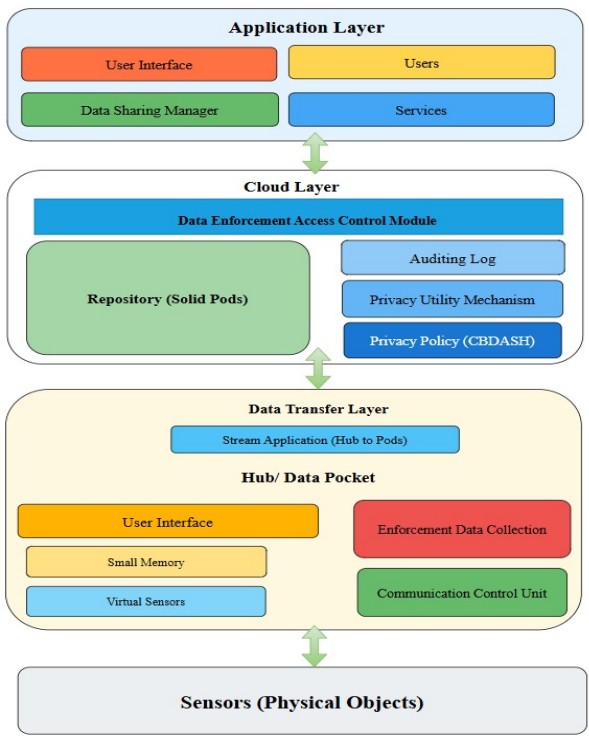

**Figure 1:** Adaptation of the Databank Architecture for Smart Homes Using Solid

example, a service that operates with energy data does not need access to health data that is stored within the same pod. And lastly an Anonymised Pod that holds partially de-identified information, which can only be read by trusted applications. This type of organisation ensures that only data in accordance with their approved groupings can be accessed by the applications, and this is beneficial in ensuring compliance with locally defined privacy policies. It assists in storing real-time and periodic IoT streams without exposing the raw streams to the cloud. The architecture ensures that there is a separation of duties. The hub is in charge of data collection and data filtering, Solid pods oversee structured storage. Access to data is mediated by DataBank, and controlled by user-defined policies. This segregation guarantees uniformity in policy implementation and limits contact with the outside servers. Pre-filtering also reduces the volume of data sent to the cloud, depending on device activity and category rules, which helps with efficiency. The system also eliminates trust in the cloud provider by restricting application access to Solid pods, giving users direct control over applications.

## 4. Databank Architecture on Solid

Access control in the Smart Home Databank assumes that a category-based policy already exists. In this paper, we follow the simplified CBDASH model, which specifies a set of data categories and application categories together with high-level rules that relate them [2]. Categories may be defined manually by users or by external processes. DataBank ensures that the policy is consistently enforced across storage and access mechanisms [14]. The policy contains all information needed for enforcement, including the categorisation of data and application and the rules that specify which application categories are allowed to access which data categories. This component represents the user's privacy preferences in a structured form and acts as the authoritative source for access control decisions.

The Databank architecture on Solid that we propose accommodates two distinct access control enforcement models. In both models, data is organised into categories and stored into corresponding pods. The differences between the approaches are in the component where access control decisions are enforced and in the interface for application services (in one model application services interact directly

with Pods, and in the other through the Databank). The distinction lies in whether Solid is exposed as the access interface or encapsulated entirely within the Databank. In the first model, access control is Pod-mediated. Data is kept in category-specific pods. However, Solid provides the primary interface for data access so applications can access the pods directly. Each pod enforces access control locally by using a predefined static Solid access control policy. As a result, the range of policies that may be implemented in the Solid-mediated approach is more restricted, due to expressiveness limitations and manual requirement listing of authorised applications. These policies specify which types of apps can access the data contained in each pod. When an application request access to the pod, Solid compares the request directly to the pod's access control policy. In this model, access control decisions are distributed across pods, while the Databank remains responsible for defining categories and routing data to the appropriate storage locations. This model has limited scalability and flexibility, thus making it difficult to support richer or evolving privacy requirements. In the second model, access control is Databank-mediated. Data from different categories is kept in separate Solid pods, however these pods are not directly accessible by the application. Instead, each pod is set up with a simple and static access control policy that only allows access to the Databank, which provides the interface through which applications and users interact with data. It serves as the only enforcement point and receives access requests from applications. It uses the CBDASH policy to evaluate the application's category against the data category upon receiving the request. If the request is approved, the Databank extracts data from the relevant category-specific pod and returns it to the application. In this model category separation is preserved at the storage level, but the access control decisions are enforced centrally by the Databank. Both models use a specific system component to automatically transport approved data streams from the hub to the appropriate Solid pods. The choice between these models is based on deployment needs, trust assumptions, and the desired balance of centralised control and distributed enforcement.

## 5. Future Work

While Solid enforces explicit control policies, it is still difficult for non-technical users. According to earlier research, defining data collecting and sharing policies in cloud-IoT environments is a key usability barrier [13]. Techniques such as privacy profiles created with little user input could be investigated to facilitate category-based policy in Solid. Research on mining and generating attribute-based access control policies via categorisation has shown that policies can be formed automatically from existing system information [20, 21]. Similar techniques can be developed for solid access control policies, minimising manual application listing. A further limitation lies in the Solid-mediated approach, as it is constrained by the lack of support for category-based policies. We can aim for incorporating richer category-based policy primitives for more expressive pod-level access control to be enforced directly.

## 6. Conclusion

This paper proposes a Smart Home Databank architecture in which category-based policies control access to IoT data, with Solid pods serving as the underlying storage layer. It distinguishes between high-level reasoning and low-level access control configurations by defining privacy requirements in terms of CBDASH data and application categories and then projecting these decisions onto pod-level authorisations. This approach was shown by two different deployment options: a Databank-mediated model, where the Databank wraps Solid pods and only it can access and enforce data-sharing policies, and a Solid-mediated model, where the application services can access data in pods, and the pods policies will be the ones enforcing the data-sharing policies (but they are static and predefined; therefore, they will be more limited). Taken together, these indicate that the integration of category-based models with Solid can give a flexible foundation for user-centric smart home data management.

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
