# OpenReview forum: "Privacy-Preserving Data Sharing in Cloud IoT on Solid"
_SolidProject.org/SoSy/2026/Privacy_Session — SoSy2026-Privacy Paper_

### Official Review · ~Thanassis_Tiropanis1 · 2026-02-27
**Privacy-Preserving Data Sharing in Cloud IoT on Solid**

**Rating:** 8
**Confidence:** 4

**Review:**

The paper addresses a very relevant problem, of how other forms of access control - in this case category-based access control - can be used in decentralised environments based on Solid.

It would be good if the authors could focus a little more on the following issues so that the contribution of this work become clearer:
- There could be discussion on what are the advantages of using category-based access control and perhaps provide examples to show its strengths in environments with IoT data.
- In the Databank-mediated model where solid is used only for storage it would be good to discuss whether/how privacy preservation is accomplished if data need to be moved from the Solid pods to the Data Sharing manager. Also, what WebIDs would each of the components shown in the figure would be using?
- It would be helpful to have an example of how a category access control policy is supported on the different layers of the diagram that is provided.

---

### Official Review · ~Wout_Slabbinck1 · 2026-03-02
**Review on Privacy-Preserving Data Sharing in Cloud IoT on Solid**

**Rating:** 6
**Confidence:** 4

**Review:**

The short paper is relevant to the solid symposium as they argue that in IoT there is excessive data collection that requires proper privacy and data protection.
They highlight that currently, such policies are lengthy and complex and that capturing user preferences to enforceable policies is a difficult endeavour.
While the topic is timely, the added value of integrating Solid is not sufficiently demonstrated. Different proposals already exist for Solid to aid making categories for certain data such as type indexes and the solid application interoperability (SAI) specification.
It remains unclear what concrete benefit the Databank integration provide, especially in the deployment model where Solid is abstracted away from end-users and applications.
Finally, the authors focus on Category Based Access Control to simplify access control. Though, throughout the paper I have not seen a demonstration of how this actually is simpler than plain WAC or ACP.
An example or direct comparison in the perspective of the Resource Owner might have made a stronger case why this indeed simplifies data management.


More in depth review:
They propose an architecture where they integrate Solid within a four-layer Databank architecture.

The Databank architecture consists of four layers
- Application Layer (interfaces include RO managing privacy preferences)
- Cloud layer (handles access control enforcement, auditing, storage, contains a policy language and a privacy utility mechanism)
- Data Pocket Layer (connects with Iot Devices)
- IoT device layer

Databank itself uses in the cloud layer Category based Access Control model using the CBDASH.
Unfortunately, it was not immediately clear that CBDASH was the policy language employed.
Since this paper puts a lot of emphasis on the policy languages, an example of such a policy would help understanding.

The authors propose to use Solid as the storage component within the Cloud layer.

In the background of Solid there a couple of aspects which could be misinterpreted:
-  "WebIDs, ...  are used as the primary identifier for users and application"
	- Actually they are used to identify entities on the web. Application are identified using Client IDs, which is elaborated in section 5 of OIDC: https://solidproject.org/TR/oidc#clientids
- When talking about the dereferencing of WebIDS, the author state that "WebID is dereferenced, it returns a RDF document that describes that agent, which is then utilised in authentication and access control"
	- During authentication using Solid-OIDC, the derefencing is indeed important. However, when doing the access control assessment, only the webID (when verified) itself is used. Thus this could be misinterpreted by a reader of this paper
- on WAC the authors claim " Because of its coupling between policies and resources this limits the policy reuse and increases complexity in large-scale environments"
	- A note on this is that ACP is also tightly coupled to resources. Though both access control mechanisms do not require an access control resource per actual LDP Resource. WAC has for example `acl:default` to indicate that all resources under a container (which works recursive) use the same permissions.

In section 3, the authors suddenly talk about **anonymised data**.
This comes rather sudden as this was not elaborated in the introduction, nor in the Conclusion. Nor is there a anonymisation u nit within the Data Transfer layer, i.e the Hub.

Another issue that has been highlighted by De Mulder et al [1] is that it might be difficult to have proper anonymisation due to linkability inherently present within Solid.

The authors mention in section 3 that "Access to data is mediated by DataBank, and controlled by user-defined policies"

Later in section 4, they state something different: there are two distinct access control enforcement models:
- One were the Solid protocol is directly employed and responsible for the access control enforcement and data interface.
- Another where the Databank interface is responsible for the Category based Data Access enforcement and access.

At the end they argue "The choice between these models is based on deployment needs,
trust assumptions, and the desired balance of centralised control and distributed enforcement."

In the second mode, end-users and applications would not even know that Solid was used.
Why Solid was then employed in that scenario? What are the benefits then?


I would advise the authors to do a final grammar check for the camera ready version if it is accepted.
Here is a non-exhaustive list of some of the errors found in the submission:
Do a grammar check as well.
- section 1.3: They are used as the primary identifier for users and application -> applications
- section 3:  separatepodd

---

### Decision · Program_Chairs · 2026-03-09

Accept (Paper)